# CT-optimal touch and chronic pain experience in Parkinson's Disease; An intervention study

**Larissa L. Meijer** [1]*, **Carla Ruis**[1,2], **Zoë A. Schielen**[1], **H. Chris Dijkerman**[1], **Maarten J. van der Smagt**[1]

**1** Utrecht University, Utrecht, The Netherlands, **2** University Medical Centre Utrecht, Utrecht, The Netherlands

* l.l.meijer@uu.nl

## Abstract

One of the most underdiagnosed and undertreated non-motor symptoms of Parkinson's Disease is chronic pain. This is generally treated with analgesics which is not always effective and can cause several side-effects. Therefore, new ways to reduce chronic pain are needed. Several experimental studies show that CT-optimal touch can reduce acute pain. However, little is known about the effect of CT-optimal touch on chronic pain. The aim of the current study is to investigate whether CT-optimal touch can reduce the chronic pain experience in Parkinson patients. In this intervention study, 17 Parkinson patients underwent three conditions; no touch, CT-optimal touch and CT non-optimal touch with a duration of one week each. During each touch week, participants received touch from their partners twice a day for 15 minutes. Results show that both types of touch ameliorate the chronic pain experience. Furthermore, it appears that it is slightly more beneficial to apply CT-optimal touch also because it is perceived as more pleasant. Therefore, we argue that CT-optimal touch might be used when immediate pain relief is needed. Importantly, this study shows that CT-optimal touch can reduce chronic pain in Parkinson's Disease and can be administered by a partner which makes it feasible to implement CT-optimal touch as daily routine.

## Introduction

In Parkinson's Disease (PD) 30–85% of the patients suffers from chronic pain, one of the most underdiagnosed and undertreated non-motor symptoms of PD [1–3]. Chronic pain is defined as ongoing disabling pain which often results in reduced well-being and a lower quality of life [4]. The most commonly reported form of pain in PD is musculoskeletal, which is mostly treated by analgesics [2]. However, analgesia can have several unpleasant side effects such as nausea, headaches, constipation, confusion and memory problems [5]. Furthermore, PD patients who are treated with medication do not always report a decrease in discomfort [6]. In order to develop novel interventions to reduce chronic pain in PD, it is important to understand its underlying mechanisms.

When a noxious stimulus innervates the skin (of healthy individuals), the pain signal is processed through two systems: the lateral and the medial pain system [7]. The *lateral* system is

**Data Availability Statement:** Fully anonymized data generated during and/or analyzed during the

current study are available in the YODA repository, DOI: 10.24416/UU01-KZ91OM.

**Funding:** Funding was obtained by Meijer, Dijkerman and Smagt. There is no Grant number. The Dutch Stichting Parkinsonfonds offered the Grant. https://www.parkinsonfonds.nl/. The funders had no role in study design, data collection and analysis, decision to publish, or preparation of the manuscript.

**Competing interests:** The authors have declared that no competing interests exist.

involved in the sensory and discriminative aspects of pain, which represents the pain threshold (the minimum level at which a stimulus is perceived as painful) [8]. The *medial* system is crucial for the affective/motivational aspect of pain, which is associated with pain tolerance (the maximum level at which a pain stimulus is tolerated) [9]. In PD both the lateral and medial pain system appear to be overactive, which results in a lowered pain threshold and pain tolerance [6, 10, 11]. This means that PD patients experience pain more severely than individuals without PD [11]. Moreover, research shows that not only central pain processing systems are disrupted, but that PD also causes changes in peripheral pain transmission [1, 6]. These changes in central and peripheral pain processing complicate current pain management [10].

Interestingly, recent research suggests that CT-optimal touch might be a novel non-pharmacological alternative to alleviate pain. CT-optimal touch is a gentle stroking of the skin, which activates the small unmyelinated C-Tactile (CT) afferent nerves [12]. This tactile system can be activated by stroking between 1–10 cm/s, optimal speed is around 3 cm/s, with a soft brush or hand [13, 14]. As this type of touch can also elicit a pleasant sensation it is also referred to as affective touch [15]. Interestingly, recent behavioral research shows that CT-optimal touch can reduce pain experience in healthy individuals [16–18]. These behavioral findings can be explained by a novel model suggesting that the CT-afferent system interacts with the medial pain system at different levels of the central nervous system [19]. In particular, it may inhibit pain signals at the level of the dorsal horn of the spinal cord and as a consequence prevent higher order processing of the pain signal [20]. In addition, it may downregulate several cortical areas such as the insula and anterior cingulate cortex, which are important for the affective aspects of pain experience [16, 21, 22].

However, the above-mentioned studies have focused on acute pain experience in an experimental setting including healthy individuals. Currently, little is known about the effect of CT-optimal touch in people suffering from chronic pain [23]. One study shows that after 11 minutes of CT-optimal touch, chronic pain significantly reduced in patients suffering from primary chronic pain, secondary musculoskeletal pain and neuropathic pain [24]. As the majority of PD patients suffer from musculoskeletal pain, CT-optimal touch might reduce chronic pain in PD patients as well.

In the current 3-week intervention study, participants diagnosed with PD and suffering from chronic pain report their pain experience before the intervention and during a CT-optimal touch *as well as* a CT non-optimal touch intervention. Based on previous studies we hypothesize that CT-optimal touch may reduce chronic pain in PD patients, and to a larger extent than CT non-optimal touch. Touch will be administered by the partner, as this enhances the positive effect of CT-optimal touch on pain [22]. We can hereby also explore whether implementing CT-optimal touch in daily life and longitudinal administration is feasible. In addition to the effect of CT-optimal touch on chronic pain, the pleasantness of this type of touch will also be recorded. Based on the study of Kass-Iliyya, Leung [25], which investigated CT-optimal touch perception in Parkinson patients, we expect that CT-optimal touch is perceived as more pleasant than CT non-optimal touch. Furthermore, we will investigate whether there is a relationship between pleasantness ratings and the relieving effect of CT-optimal touch. If so, this might indicate that the pain-relieving effect is merely related to having a concurrent pleasant sensation, rather than the specific activation of the CT-system. However, previous studies of von Mohr, Krahé [22] and Meijer, Schielen [26] into the effect CT-optimal touch on respectively acute pain and itch have failed to show such a relationship between the perceived pleasantness of CT-optimal touch and its relieving effect, suggesting it is not the experienced pleasantness per se that causes the pain reduction. Finally, if CT-optimal touch can ameliorate pain this might also positively influence mood and affect [27].

## Materials and methods

### Participants

Participants who experience chronic pain and suffer from PD were recruited through ParkinsonNEXT, an online platform connecting researchers with aspiring participants diagnosed with PD (https://www.parkinsonnext.nl). Recruitment took place from September 2020 until December 2022. A total of 57 participants signed up for the study of which 31 were eligible for participation. Inclusion criteria were; age ≥18, PD diagnosis, pain associated with PD (musculoskeletal, dystonic, akathisia) and/or pain worsened by PD (i.e. (osteo)arthritis, other age-related pain conditions), pain present for at least 3 months, with clear impact on physical/psychological functioning (measured with the King's Parkinson's Disease Pain Scale), which must be assessed as at least moderate in intensity (≥4 points on an 11-point Likert pain scale) and the ability to provide informed consent. Exclusion criteria were incapability of giving informed consent, inability to understand questionnaires, suffering from conditions that affect the ability to feel or process touch, pain conditions that can also influence the perception and processing of touch; i.e. neuropathic pain, a history of cerebral traumata or psychiatric disorders unrelated to PD and currently suffering from a mood disorder. During the study 12 participants dropped out because: during the first week of the study they did not report to experience pain (N = 4); it was to difficult to combine this 3-week intervention with their daily-working-schedules (N = 5); their partner was not able to provide touch frequently due to physical limitations (N = 2). One participant dropped out because of a painful sensation elicited by CT non-optimal touch. As a result, 19 participants (8 women), aged between 31 and 76 years (M = 65.47, SD = 11.22) and mostly suffering from musculoskeletal pain successfully participated in the study (for more descriptives see S1 Table in S1 File). In addition, one participant was excluded as technical problems with Gorilla (an online survey tool) resulted in too many missing and invalid data.

One other participant was excluded from the sample. During the intervention it became plausible that this participant mainly suffered from neuropathic pain, which was one of the exclusion criteria. As this was not formally diagnosed at first, it was decided that the participant was eligible for participation. This participant's chronic pain appeared fully diminished after the end of the study, and such an effect of CT-optimal touch had not been reported by any of the other participants from this sample. Therefore, the results of this participant are reported in a single case report [28] and this patient was not included in the current study. Thus, data of 17 participants was used for the analyses. All participants provided written informed consent, which were stored separately from all anonymized data. The study was performed in agreement with the Declaration of Helsinki and has been approved by the medical research ethics committee at the UMC Utrecht (NL71563.041.20).

### Design

This was a 3-week intervention study with a within-subjects design which included a baseline condition ('no touch' week) and two experimental conditions (CT-optimal touch and CT non-optimal touch). The order of the experimental conditions was counterbalanced across participants. The primary outcome measure was the subjective pain experience measured with the Color Analogue Scale (CAS) and Faces Pain Scale-Revised (FPS-R).

### Procedure

After an aspiring participant signed up through ParkinsonNEXT, contact information was sent to the experimenter. When a participant was interested in participating, had read the

information letter and met the inclusion criteria, the experimenter made an appointment. During the first appointment the experimenter provided information regarding the procedure, the Montreal Cognitive Assessment (MoCa) or Telephone Interview for Cognitive status Modified (TICS-M) and Pain Intensity Scale (PIS)were filled out, informed consent was given and hereafter the Kings Parkinson Disease Pain Scale (KPDPS) and Quality of Relationship Inventory (QRI) were filled out. Hereafter, the experimenter demonstrated the two types of touch and asked the partner to apply touch so the experimenter could check whether the instructions were clear. The experienced pleasantness of both types of touch was also measured by asking the patient to rate touch on a scale from 0 to 10, to control for possible unpleasant sensations elicited by either type of touch.

During the study, all participants partook in the following sequence: one week of pain registrations only (no-touch), one week of CT-optimal touch (and pain registrations) and one week of CT non-optimal touch (and pain registrations) (see Fig 1 for an outline of the procedure). Upon completion of the experiment, or when participants preliminary withdrew from the experiment, they were debriefed.

**One week of pain registrations only ('no touch' week).** Starting the day after the pre-intervention baseline measurement, participants reported their pain experience, measured by the CAS and the FPS-R, three times a day for one week to control for normally present pain fluctuation. The questionnaires were provided by Gorilla Experiment Builder (www.gorilla.sc), a survey tool. Two participants were not able to use Gorilla and were therefore provided with a hardcopy of the questionnaires.

**One week of CT-optimal touch/CT non-optimal touch.** The procedure for both touch weeks was identical. Before the start of the touch week the experimenter video-called the participant during which the type of touch was demonstrated again, and the procedure was explained. The next day the participant and partner started with one week of touch stimulation. Touch was administered twice a day (morning and evening, 15 min) by the partner of the patient. Pain ratings (CAS Pain Scale and FPS-R) were measured before (at 0 min), during (at 5 and 10 min) and after the stimulation (at 15 min). Partners kept track of time by using a (stop)watch. In addition, the pleasantness of the stimulation was also measured during (at 5 and 10 min) and after touch stimulation (at 15 min) with the VAS pleasantness scale. To assess whether there were any long(er) term effects of touch stimulation during the day, pain ratings (CAS Pain Scale and FPS-R) were also measured in the afternoon. Halfway through the week, the experimenter video-called the participant and partner to ensure uniform touch administration. The partner was asked to perform the touch without any prior instruction, so that the experimenter could check whether touch was still applied correctly. This call was recorded, anonimized and touch performance of most of the participants was also checked by the other research team members. On the first and last day of the touch week participants filled out the Positive and Negative Affect Scale (PANAS).

## Materials and measures

Several questionnaires were administered before and during the study. The questionnaires are categorized based on the moment of administration.

**Before the start of the study.** *Pain intensity*. The Pain Intensity Scale (PIS) was used to measure the intensity of the experienced chronic pain at that particular moment; the participant filled out a 11-point Likert pain scale for pain intensity. A score of <4 points was used as an exclusion criterion.

*Chronic pain*. The participants' chronic pain experience was measured using the Kings Parkinson's Disease Pain Scale (KPDPS). This was used as a baseline pain measurement. The

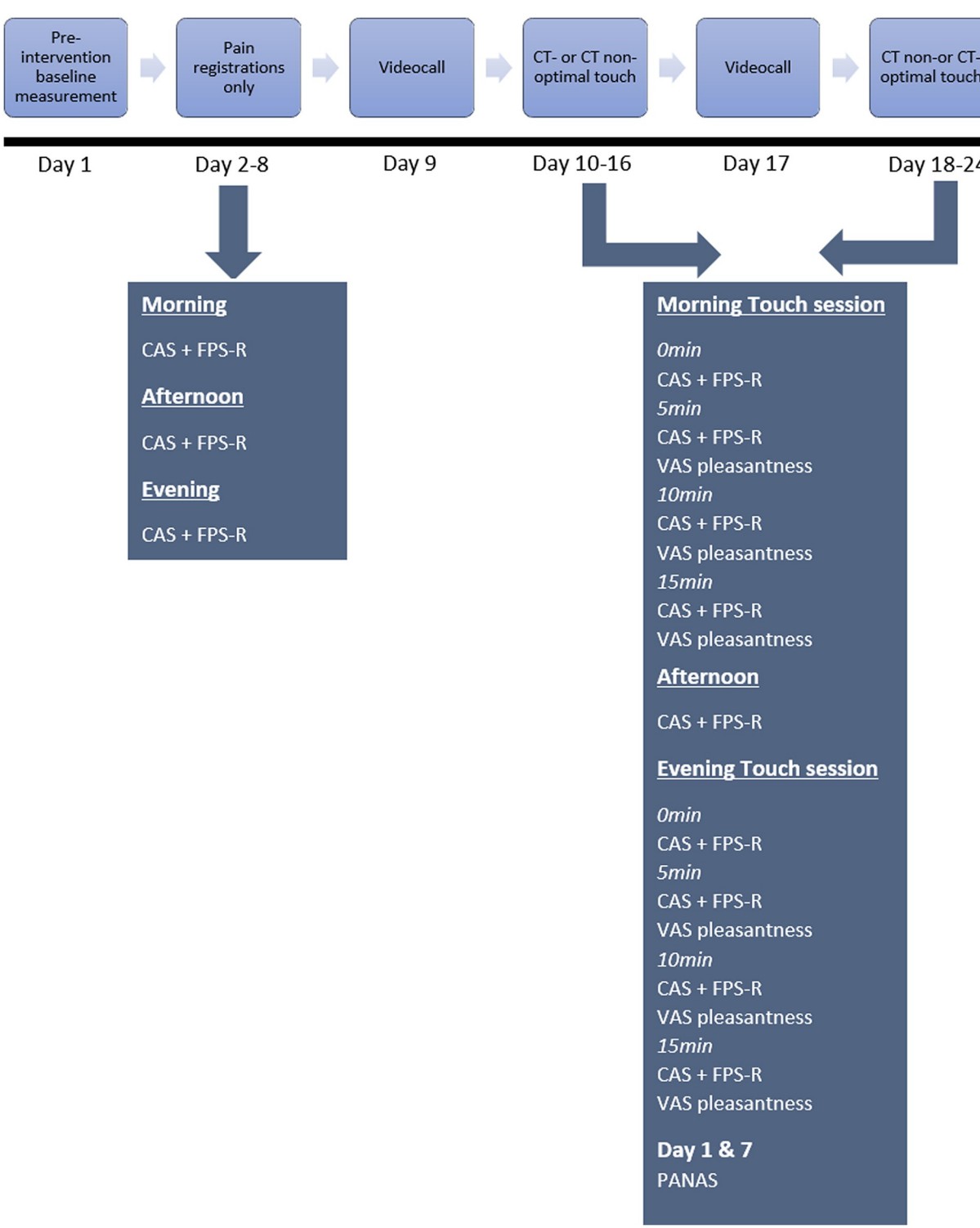

**Fig 1. Outline of the intervention.**

KPDPS measures the intensity and severity of pain as well as localization and its relationship with motor fluctuations or musculoskeletal pain. This is an interviewer-based questionnaire with 7 domains which are based on common types of chronic pain in Parkinson patients. The total score is the sum of all domains, which are based on the severity multiplied by frequency [29].

*Cognition*. To assess whether participants were able to provide informed consent the Montreal Cognitive Assessment (MoCa) or Telephone Interview for Cognitive status Modified (TICS-M) was administered. Research shows that in PD patients, the MoCa can detect the likelihood of impaired cognitive capacity in which a cut-off score of ≤22 is most sensitive (94%) [30]. As face-to-face testing was not always feasible due to the COVID-19 pandemic, the TICS-M was also used to assess the ability to provide informed consent. The TICS reflects general cognitive ability and can detect cognitive impairments, a cut-off score of <34 was used as this might indicate mild cognitive impairment [31].

*Quality of relationship*. As the partner of the participant provided touch during the study, we wanted to check whether there was no discrepancy between perceived quality of the relationship of the participant and their partner, as this might influence the way touch was applied and/or how touch was perceived. The quality of the relationship in terms of perceived support of the participant and partner was assessed through the Quality of Relationships Inventory (QRI) short form [32].

**During the study.** *Pain intensity*. The Colour Analogue Scale (CAS) was used to measure pain *intensity*. The CAS is a questionnaire that measures pain intensity by using different colours: the white-coloured bottom represents 'no pain' and the dark red top represents 'extreme pain' (see Fig 2). These colours are linked to a numeric scale from 0–10 which is not visible for the participant [33]. In this study the CAS was used digitally (computer, phone or tablet) by clicking with a mouse or finger (touch screen) on a point in the scale which represent the current pain intensity.

*Pain severity*. The Faces Pain Scale-Revised (FPS-R) was used to measure the *severity* of pain and the affective component. The FPS-R that contains six faces, on the left a neutral face and moving to the right five faces which express increasing feelings of pain (see Fig 3). The neutral face represents 0 'no pain' and the five painful faces represent an ascending score of 2, 4, 6, 8, 10 of which the latter represent 'severe pain' [33, 35, 36]. The FPS-R was used digitally by clicking (or touching) on the face representing the participants pain severity.

*Pleasantness*. The pleasantness of both types of touch was registered by a Visual Analogue Scale (VAS) ranging from 0–10, in which 0 represented 'unpleasant' and 10 'pleasant'.

*Mood/affect*. To assess the two dimensions of mood, namely positive- and negative affect, the Positive and Negative Affect Scale (PANAS) was used. This is a 20-item questionnaire, which has been shown to be a reliable, valid and efficient measure for positive- and negative affect [37]. The PANAS was provided before (day 1)- and after (day 7) each touch intervention week, to measure if touch also influenced the participants affect in general.

*Other*. Participants received a diary in which relevant information could be reported, this included usage of pain medication, changes in daily activities which might influence pain and changes in quality of sleep.

**Tactile stimulation.** Two types of touch were administered to the participant. CT-optimal touch was administered by the participant's partner by stroking the dorsal forearm of the participant with the hand at a slow but natural speed of around 3 cm/s. This was done by moving from elbow to wrist in approximately 6 seconds. As a control condition CT non-optimal touch was administered by stroking the forearm at a faster but still natural speed of around 18 cm/s. This was done by moving from elbow to wrist in approximately 1 second. Partners received a demonstration and an instruction sheet on how to apply the type of touch. In addition, they

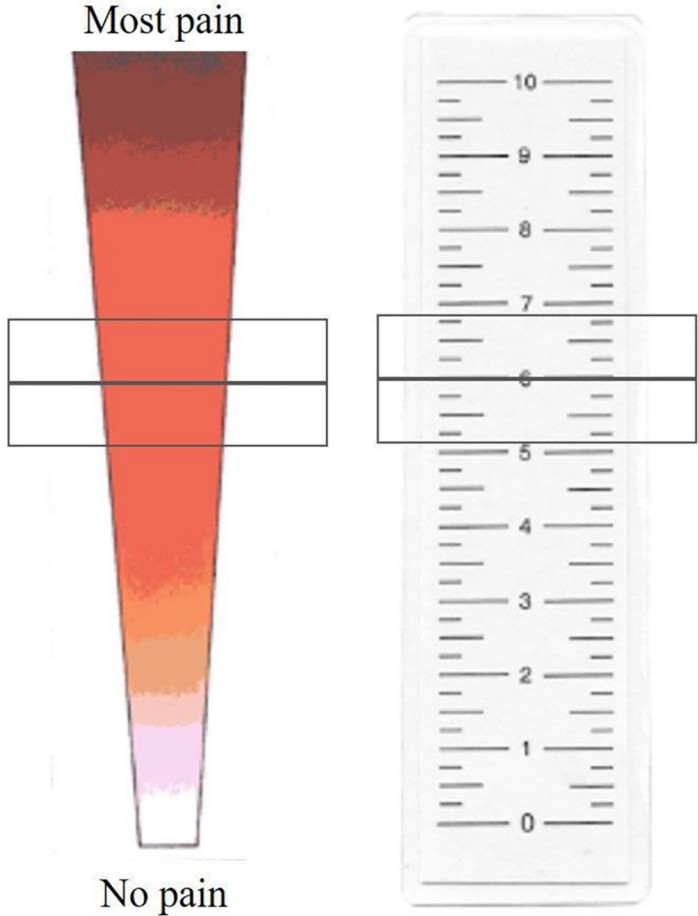

**Fig 2. Color Analogue Scale.** Adapted from [34].

also received a video in which the touch was demonstrated so they were able to look back and consult the demonstration at any time.

## Statistical analysis

All data was processed using Microsoft Excel (version 2208) and analysed with SPSS Statistics (version 28). Due to technical problems some participants unfortunately had trouble with

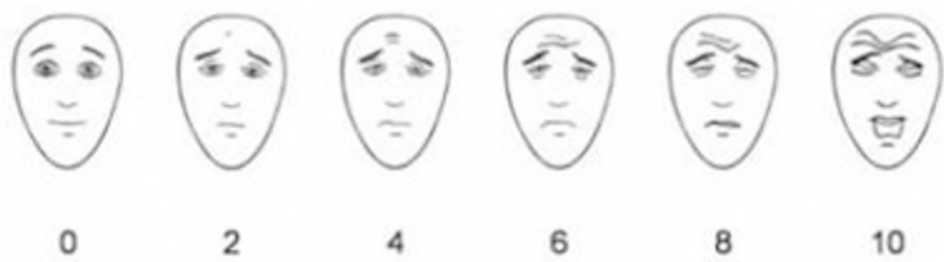

**Fig 3. Faces Pain Scale-Revised.** Adapted from [35].

reporting their pain experience in Gorilla, especially with the CAS Pain Scale. Therefore, the data of the CAS Pain Scale was deemed invalid and excluded from analyses. As we do not know from previous studies when and how possible pain reduction through CT-optimal touch compared to CT non-optimal touch may occur, several analyses were done. An apriori power calculation for a repeated measures ANOVA with expected power (.80), medium effect size Cohen's F (.25) and alpha (.05) recommend a sample size of 24. However, the observed power for the main analysis is .995 which reflects high statistical power for N = 17.

Before analyzing, several steps were taken to process the data. First, data was retrieved from Gorilla and processed in Excel by creating different tabs for the no touch, CT-optimal touch and CT non-optimal touch condition. Second, to analyze the 'overall-effect' of the touch intervention, averages of the FPS-R data for day 1 –day 7 were calculated. For the no touch week this means the average of the morning, afternoon and evening data points. For the CT-optimal touch and CT non-optimal touch condition this means the average of the morning and evening data when touch was administered and the afternoon data points when no touch was administered. This data was analyzed in SPSS with a 3 (no touch, CT-optimal touch and CT non-optimal touch) x 7 (days) repeated measures ANOVA. Data was normally distributed, sphericity was violated for the touch x day interaction effect therefore Greenhouse-Geisser corrections were used. A Bonferroni post-hoc comparison was used to analyze the difference between the three conditions.

To analyze whether there is a difference between the pain ameliorating effect of CT-optimal touch and CT non-optimal touch during the stroking, i.e., 'short-term effect', data from the different timepoints (0min, 5min, 10min, 15min) were used. An average day score for every timepoint was calculated for both types of touch. This data was then analyzed in SPSS with a 2 (CT-optimal touch and CT non-optimal touch) x 7 (days) x 4 (0min, 5min, 10min, 15min) repeated measures ANOVA. Data was normally distributed, sphericity was mildly violated for some factors therefore the Greenhouse-Geisser correction was used.

The pleasantness of both types of touch was calculated by averaging the VAS data points on 5min, 10min and 15min for morning and evening; this was done for every day. This data was analyzed in SPSS with a 2 (CT-optimal touch and CT non-optimal touch) x 7 (days) repeated measures ANOVA. Not all data was normally distributed according to the Shapiro–Wilk test. As a non-parametric alternative for a factorial ANOVA is not readily available and it has been shown that Type 1 error and power of the F-statistic are not necessarily altered by violation of normality [38], we decided that it was permitted to use a parametric test. Sphericity was violated, therefore Greenhouse-Geisser corrections were used.

For the PANAS scores a positive- and negative affect score was calculated. For the positive affect score, the scores on items 1, 3, 5, 9, 10, 12, 14, 16, 17, 19 were added. For the negative affect score, the scores on items 2, 4, 6, 7, 8, 11, 13, 15, 18, 20 were added. This was done for day 1 and day 7 and for CT-optimal touch and CT non-optimal touch condition. The PANAS score of both types of touch were analyzed with a 2 (CT-optimal and CT non-optimal touch) x 2 (positive- and negative effect) x 2 (days) repeated measures ANOVA.

To analyze the relation between the pain ameliorating effect of touch and perceived pleasantness a FPS-R difference score was calculated. First, an average score for 0min and 15min was calculated for every day. The FPS-R difference score was calculated by substracting 0min from 15min. This was done for every day, whereafter data over the week was calculated by averaging day 1 –day 7. The average VAS pleasantness score was used as a measure for perceived pleasantness. Which was than analyzed with a Spearman correlation. In addition, to analyze the relationship between perceived pleasantness and quality of the relationship (measured with the QRI), first a QRI total score was calculated for the participant and the partner. A difference score was then calculated by substracting the partner score from the participant

score. A Spearman correlation was used to analyze the relation between the VAS pleasantness and QRI difference score.

## Results

Before the start of the study participants' average PIS was 5.85 (SD = 1.55) and for the KPDPS the average score was 38.06 (SD = 15.43). The participants' total score on the QRI was 16.06 (SD = 1.95) and the partner's score was 15.38 (1.80). The information reported in the diary was checked by the experimenter after participants finished the study. No changes or particularities were reported. To provide a clear-overview of the collected data and used analyses, outcomes are described as the overall-effect, short-term effect including optimal touch duration, pleasantness, affect, relationship between pleasantness and short-term effect and relationship between pleasantness and quality of the relationship.

### Overall- effect

The average FPS-R pain day scores for the three conditions NT, CT- optimal touch and CT non-optimal touch were analyzed with a repeated measures ANOVA, which showed a significant main effect of touch F(2,32) = 14.28, p < .001, partial $\eta^2$ = .47 and observed power of .995. There was a significant effect for day F(6,96) = 3.05, p = .016, partial $\eta^2$ = .16. There was no significant interaction between touch x day (p = .061). A bonferonni corrected post-hoc comparison showed a significant effect between NT–CT-optimal touch (p < .001) and NT–CT non-optimal touch (p = .003). There was no significant difference between CT-optimal touch–CT non-optimal touch (p = 1.00). Thus, the FPS-R scores were significantly lower for the CT-optimal touch and CT non-optimal touch conditions compared to the no-touch condition (see Table 1 and Fig 4).

As the type of touch was counterbalanced between participants, the data was checked for a possible order effect. An average week FPS-R score was calculated for every condition and hereafter a difference score was calculated by substracting the CT-optimal touch and CT non-optimal touch week from the NT week. A touch (2 levels) x order (2 levels) repeated measures ANOVA was used to analyze the data. There was no significant difference between NT minus CT-optimal touch and NT minus CT non-optimal touch over the week F(1,15) = .48, p = .499. There was also no significant interaction effect between touch x order F(1,15) = .15, p = .702.

### Short-term effect

To analyze the difference between touch conditions and the different timepoints, first for both touch conditions an average FPS-R score for every timepoint was calculated per day. The latter was done to also analyze whether there are any differences over the 7 days (see Table 2 and Fig 5).

**Table 1. Mean (SD) pain scores on the FPS-R per condition over the days.**

|  | No touch | CT-optimal touch | CT non-optimal touch |
|---|---|---|---|
| Day 1 | 4.63 (1.42) | 3.39 (1.67) | 3.51 (1.65) |
| Day 2 | 4.12 (1.69) | 2.90 (1.38) | 2.93 (1.40) |
| Day 3 | 5.41 (2.03) | 3.19 (1.61) | 3.14 (1.35) |
| Day 4 | 4.00 (1.68) | 3.89 (1.17) | 3.42 (1.61) |
| Day 5 | 3.49 (1.40) | 2.99 (1.20) | 3.02 (1.68) |
| Day 6 | 4.08 (1.84) | 3.29 (1.29) | 3.39 (1.99) |
| Day 7 | 3.88 (1.46) | 3.27 (1.45) | 3.31 (1.56) |

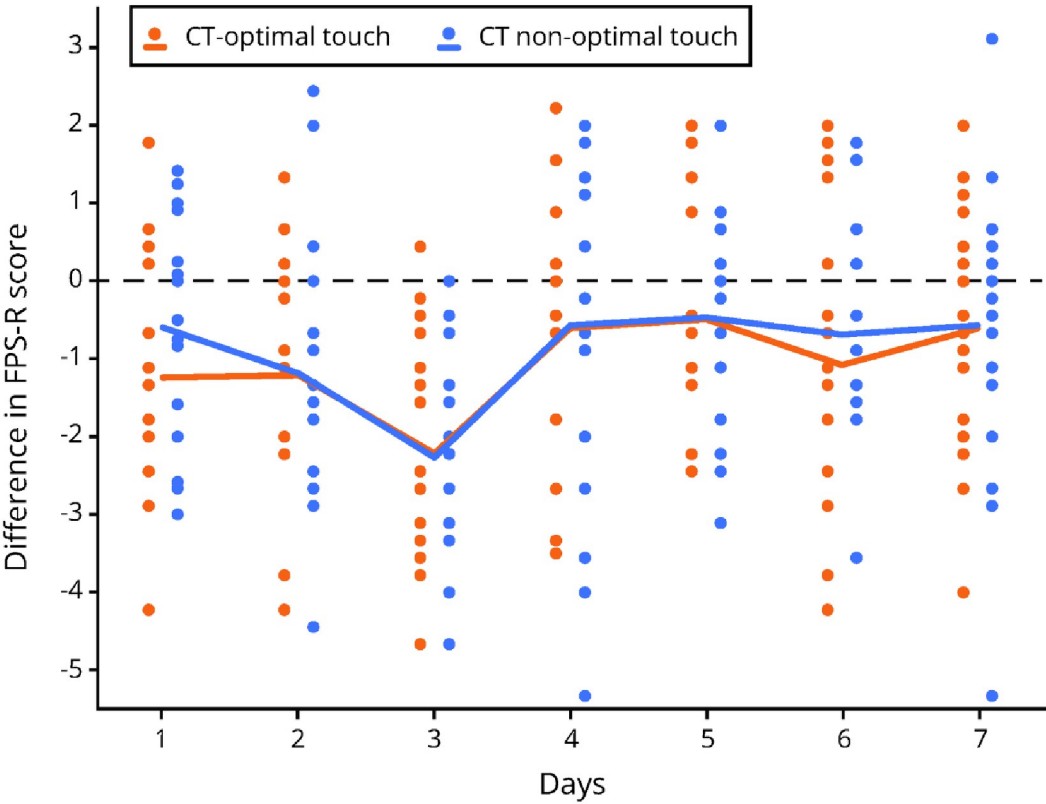

**Fig 4. Scatterplot depicting FPS-R difference scores per day for CT-optimal touch—NT and CT non-optimal touch—NT.** The dots represent individual datapoints, the lines show the sample mean.

A touch (2 levels) x day (7 levels) x timepoint (4 levels) repeated measures ANOVA was done to analyze the difference between timepoints for CT-optimal touch and CT non-optimal touch. There was no significant main effect for type of touch $F(1,16) = .10$, $p = .760$ or for day $F(6,96) = 1.57$, $p = .191$. There was a significant main effect for timepoint $F(3,48) = 40,77$, $p < .001$, partial $\eta^2 = .72$. There was also a significant interaction between touch x timepoint $F(3,48) = 3.75$, $p = .033$, partial $\eta^2 = .19$, indicating a slightly larger ameliorating effect for CT-optimal touch compared to CT non-optimal touch. There was no significant interaction between touch x day ($p = .975$) or day x timepoint ($p = .495$) nor for touch x day x timepoint ($p = .693$). A bonferonni corrected post-hoc comparison was used to analyze the difference between time-points for CT-optimal touch and CT non-optimal touch. For CT-optimal touch there is a significant difference between 5min– 0min ($p = .005$), 10min– 0min ($p < .001$), 15min– 0min ($p < .001$), 10min– 5min ($p = .001$) and 15min– 5min ($p < .001$), but not

**Table 2. FPS-R scores (mean (SD)) for CT-optimal touch and CT non-optimal touch for the different timepoints.**

|       | CT-optimal touch | CT non-optimal touch |
|-------|------------------|----------------------|
| 0min  | 3.61 (1.16)      | 3.53 (1.31)          |
| 5min  | 3.25 (1.13)      | 3.28 (1.39)          |
| 10min | 2.94 (1.16)      | 3.03 (1.25)          |
| 15min | 2.78 (1.13)      | 3.01 (1.31)          |

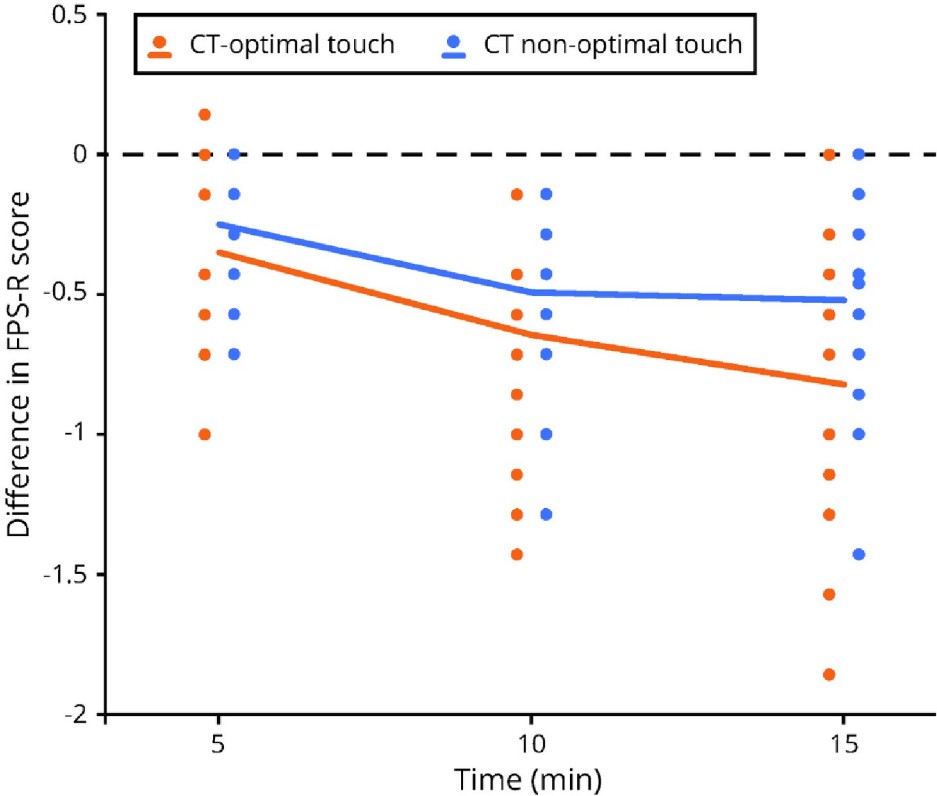

**Fig 5. Scatterplot depicting the difference in FPS-R scores during touch (5, 10, 15min) compared to no touch (0min).** The dots represent individual datapoints, the lines show the sample mean.

between 15min– 10min (p = .079). For CT non-optimal touch there is a significant difference between 5min– 0min (p = .003), 10min– 0min (p < .001), 15min– 0min (p < .001), 10min– 5min (p = .009) and 15min– 5min (p = .013), but not between and 15min– 10min (p = 1.00) (see S2 Table in S1 File).

## Pleasantness

The difference in VAS pleasantness scores for CT-optimal touch and CT non-optimal touch were analyzed with a 2 (touch) x 7 (days) repeated measures ANOVA. There was a significant main effect for type of touch $F(1,16) = 17.84$, $p < .001$, partial $\eta^2 = .53$. There was no significant main effect for day (p = .676) or for the interaction between touch x day (p = .720) (see Table 3).

## Affect

The PANAS scores at day 1 and day 7 were used to analyze if there is a difference in positive or negative affect during the CT-optimal touch and CT non-optimal touch condition (see Table 4). Several participants did not fill out (parts of) the PANAS due to technical difficulties, therefore N = 14. A touch (2 levels) x Positive or negative affect (2 levels) x day (2 levels) ANOVA was used to analyze the data. There was no significant main effect for touch $F(1,13) = .37$, $p = .555$ or day $F(1,13) = .03$, $p = .867$ and there was also no significant interaction between

**Table 3. Mean (SD) pleasantness scores for CT-optimal touch and CT non-optimal touch over the days.**

|  | CT-optimal touch | CT non-optimal touch |
|---|---|---|
| Day 1 | 7.49 (1.87) | 6.33 (1.79) |
| Day 2 | 7.82 (2.22) | 6.30 (1.93) |
| Day 3 | 7.75 (2.18) | 6.19 (2.04) |
| Day 4 | 7.81 (2.01) | 6.22 (1.89) |
| Day 5 | 7.98 (2.00) | 6.35 (1.90) |
| Day 6 | 7.85 (2.02) | 6.22 (1.96) |
| Day 7 | 7.88 (2.10) | 6.35 (2.02) |

**Table 4. The mean (SD) PANAS scores for the positive and negative affect scale on day 1 and 7.**

|  | Positive affect | Positive affect | Negative affect | Negative affect |
|---|---|---|---|---|
|  | *Day 1* | *Day 7* | *Day 1* | *Day 7* |
| CT-optimal touch | 26.79 (7.23) | 27.86 (8.21) | 17.14 (6.72) | 16.29 (4.83) |
| CT non-optimal touch | 27.36 (7.30) | 26.57 (7.28) | 17.57 (7.05) | 17.79 (7.18) |

touch x day F(1,13) = .05, p = .829. As there is no significant difference, the PANAS scores were not further analyzed.

## Pleasantness and short-term effect CT-optimal touch

To analyze whether the short-term effect of CT-optimal touch may be related to the perceived pleasantness of this type of touch, a Spearman correlation was used. No significant correlation was apparent between the short-term effect of CT-optimal touch and its perceived pleasantness, ρ = -.31, p = .228.

## Pleasantness and quality of the relationship

To analyze whether the quality of the relationship influenced how both types of touch were perceived, a Spearman correlation was used. There was no significant correlation between the QRI difference score and VAS pleasantness for CT-optimal touch (ρ = .02, p = .943) and for CT non-optimal touch (ρ = .05, p = .865). We therefore assume that quality of the relationship did not influence touch perception.

## Discussion

Previous studies have shown that CT-optimal touch can reduce acute pain in an experimental setting [17, 18, 21, 22]. However, only little is known about the effect of CT-optimal touch on chronic pain and this was tested in an experimental setting [23, 24]. As PD patients are known to suffer severely from chronic pain and currently effective treatment is missing [6], the aim of the current study was to investigate the influence of CT-optimal touch on chronic pain experience in PD in a non-experimental setting. By doing so, the feasibility of CT-optimal touch application by a partner in daily life could also be explored.

As this is, as far as we know, the first study into the effects of CT-optimal touch on chronic pain in PD, several analyses were conducted to investigate if and when pain amelioration occurs. The results show that both types of touch significantly reduce experienced pain compared to the no touch condition. That is, both CT-optimal touch and CT non-optimal touch are effective in reducing the chronic pain experience in PD, and the order of the touch

condition did not influence this result. In addition to this overall-effect, the results show (by the interaction between touch and timepoint, see also Fig 5) that CT-optimal touch appears to reduce the chronic pain experience slightly more than CT non-optimal touch.

The current results show that compared to the no touch condition, CT non-optimal touch may be almost as effective in reducing chronic pain as CT-optimal touch. Based on the study of Di Lernia, Lacerenza [24] and the underlying mechanisms of both types of touch, this was unexpected. However, in the study of Di Lernia, Lacerenza [24] the control condition was touch vibration applied by a device instead of CT non-optimal touch applied by a partner. CT non-optimal touch is a faster touch which is naturally applied with more force than CT-optimal touch, most likely activating the Aβ-fibers [22, 39]. From previous research it is known that the Aβ-fibers interact with pain on the level of the spinal cord. This is also referred to as the gate-control theory, which however has been criticized and might not fully explain the effect of CT non-optimal touch in the current study [40]. Interestingly, as touch has been applied by the partner, the social component underlying touch might have added to the pain reducing effect of CT non-optimal touch, similar to the pain reducing effect that can be observed as a result of handholding, which also appears to rely mostly on the Aβ-fibers [41]. Receiving social touch, such as caressing, handholding and massages has also been shown to have a positive influence on pain experience [42]. Another explanation might be that with a velocity of 18 cm/s the CT-afferents were activated as well. Even though microneurography studies show that the optimal velocity to activate the CT-fibers is 1–10 cm/s and faster stimuli (>10 cm/s) only activate a handful of CT-afferents, it has so far not been tested whether the CT-afferents react to 18 cm/s [43].

Interestingly, it appears that it might be slightly more beneficial to use CT-optimal touch compared to CT non-optimal touch. There is an interaction effect between touch and timepoint which reflects the steeper slope and larger difference between 0 minutes and 15 minutes for CT-optimal touch as observed in Fig 5. This is in line with previous studies into acute pain and chronic pain in an experimental setting, in which CT-optimal touch was effective in reducing pain experience [16, 22, 24].

In addition, CT-optimal touch is also rated as significantly more pleasant than CT non-optimal touch. Furthermore, at least 10 participants and their partners subjectively reported at the end of the study, before debriefing, that they were planning on continuing CT-optimal touch administration as it felt most effective in diminishing their pain and it was more natural and pleasant to receive and apply. Participants who had higher pain levels during the evening also subjectively reported that it felt that CT-optimal touch effectively reduced this "worst pain", and this also improved their quality and ability to sleep through the night. So, there appears to be a clear preference for CT-optimal touch by participants and their partners.

As CT-optimal touch is clearly perceived as more pleasant, a possible relation between the effects of CT-optimal touch on pain and perceived pleasantness is of interest. We therefore investigated whether there is a relation between the effect of CT-optimal touch and its perceived pleasantness. Since no such relation was found, the pain-relieving effect of CT-optimal touch appears to be independent of its perceived pleasantness. A similar result has been reported before by von Mohr, Krahé [22], Meijer, Schielen [26] and Meijer, Ruis [28]. From the model of Meijer, Ruis [19] it follows that CT-optimal touch may reduce pain, based on a bottom-up as well as a top-down process. As the patients in the current study were suffering from (lower) back pain and/or pain in the shoulders/neck, and touch was applied on the forearm, one could argue that it is more likely that top-down processes are involved instead of peripheral bottom-up processes. The top-down process relies on downregulation of pain regions involved in the motivational aspects of pain processing, i.e. the Insula and ACC, regions which are also highly involved in the perceived pleasantness of CT-optimal touch [44].

As we did not find a relationship between perceived pleasantness and the pain-relieving effect of CT-optimal touch, we speculate that this top-down process (i.e. downregulation through the Insula and ACC) might therefore rely on the activation of the CT-fibers instead of activation by the perceived pleasantness of touch. This would mean that the top-down influence is not necessarily a pleasantness related regulatory system but relies on input from the CT-fibers. This notion is important as chronic pain can also affect how we perceive touch, e.g. neuropathic pain, and it has therefore been suggested that CT-optimal touch might be ineffective in this patient group [45]. However, the current results indicate that how CT-optimal touch is perceived, i.e. pleasant vs unpleasant, does not influence its pain ameliorating properties, a notion also emphasized by the single case report of Meijer, Ruis [28] and the study of Di Lernia, Lacerenza [24].

The above mentioned case report of [28] describes a participant who was excluded from the participant sample in the current study. Overall, the current study and the case report both show that CT-optimal touch can reduce chronic pain experience. However, in the case report CT-optimal touch fully diminished the experienced chronic pain which has not been reported by one of the participants in the current study. This might be explained by a different chronic pain pathology [46]. The participant in the case report suffered from a burning pain in his hands possibly caused by neuropatic pain, while the participants in the current study suffered from musculoskeletal and radicular pain. Therefore, different underlying mechanisms could be involved which might also respond differently to CT-optimal touch. Another explanation might be related to touch application site. As touch was always applied on the forearm, for the patient described in the case report touch was applied on the same body part as where chronic pain was experienced, for the participants in the current study touch was applied on a different body part than where the chronic pain was experienced. As mentioned, we speculate that for the current study pain reduction through CT-optimal touch might rely more on top-down processes. For the case report on the other hand, pain reduction might rely more on bottom-up pheripheral process which in this case appears highly effective. So compared to the current study, the case report highlights that certain individuals do experience additional benefits from CT-optimal touch and therefore further longitudinal research into the effect of CT-optimal touch on chronic pain experience is warranted.

Further research into the optimal duration of applying touch is warranted as well. It appears that the effect of touch on pain experience levels off between 10–15 minutes. Therefore, we carefully conclude that to reduce chronic pain touch should be applied at least 10 minutes. This is in line with the study of Di Lernia, Lacerenza [24] in which CT-optimal touch appears effective after 11 minutes. However, as we did not investigate what happens directly after touch application has stopped, we also do not know whether a reduction in pain would still occur after 15 minutes even when touch is applied for only 5–10 minutes. Therefore, further investigation into the optimal duration of applying touch is necessary.

The current study has several strengths and limitations. A first strength is that this is the first study in which the effects of CT-optimal touch on chronic pain are studied over a more prolonged period of time. In addition, this study shows that CT-optimal touch can be used in a home-setting and application can be done by a partner who received a short and simple training. Participants and partners reported they never missed a touch application throughout the study. However, there were also 5 participants who dropped out because it was difficult to combine this intervention with their daily-working schedules. As this was merely linked to the tight schedule of this study and not to administration of CT-optimal touch, we believe that it is feasible to implement CT-optimal touch in daily life to reduce the chronic pain experience. The current study also has a number of limitations. One is that due to technical issues, resulting in problems with reporting chronic pain experience, the reliability of the data might be

affected (possibly obscuring effects). Second, we have a relatively small sample size of n = 17, which negatively influences the ability to generalize the results to a larger population. A third, important, limitation is that we did not use a control condition. We used the 'no touch' week as a control condition for every participant. As we did not use a 3-week no touch control condition we do not know whether pain experience would also have changed over time. However, as the experienced chronic pain needed to be present for at least 3 months one would not expect a significant difference over a 3-week time period without any treatment. In addition, we did not use any other form of affectionate behavior as a control condition. As mentioned before, touch by a partner has a strong social component which could explain why CT non-optimal touch also reduced chronic pain experience [42]. So, it could be that any form of affectionate behavior might have similar effects [47]. This is in line with previous studies showing that the mere presence of a partner, as a form of passive social support, can reduce pain as well [48]. Finally, as this study is a within-subject design, participants and their partners were not blinded. Even though the terms CT-optimal- or affective touch have not been used and the theoretical background was discussed with participants only during the debriefing, it could be that during the study participants became aware of the differences in presumed effectiveness of the types of touch. However, we think it unlikely this would have influenced our results as we found no significant order effect. Based on our current study, we suggest that future studies employ a Randomized Control Design and administer touch for more than one week.

To conclude, the current study investigated whether CT-optimal touch can reduce chronic pain experience in PD patients. Overall, both CT-optimal and CT non-optimal touch are effective in relieving chronic pain compared to no touch. Furthermore, it appears that it is slightly more beneficial to apply CT-optimal touch. As it is also perceived as more pleasant and reported to be more feasible to apply, we speculate that CT-optimal touch might be used when there is a need for immediate pain relief. Furthermore, our current study shows that CT-optimal touch application by the partner is feasible, which further emphasizes the possibilities of using CT-optimal touch as a treatment for chronic pain.

## Supporting information

**S1 File.**
(DOCX)

## Author Contributions

**Conceptualization:** Larissa L. Meijer, Carla Ruis, H. Chris Dijkerman, Maarten J. van der Smagt.

**Data curation:** Larissa L. Meijer, Zoë A. Schielen.

**Formal analysis:** Larissa L. Meijer, Maarten J. van der Smagt.

**Funding acquisition:** Larissa L. Meijer, H. Chris Dijkerman, Maarten J. van der Smagt.

**Methodology:** Larissa L. Meijer, Carla Ruis, H. Chris Dijkerman, Maarten J. van der Smagt.

**Supervision:** Carla Ruis, H. Chris Dijkerman, Maarten J. van der Smagt.

**Writing – original draft:** Larissa L. Meijer, Carla Ruis, Zoë A. Schielen, H. Chris Dijkerman, Maarten J. van der Smagt.

**Writing – review & editing:** Larissa L. Meijer, Carla Ruis, H. Chris Dijkerman, Maarten J. van der Smagt.

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
