## [Decision Letter · Decision Letter 0]

22 Aug 2023

PONE-D-23-21643Affective touch and chronic pain experience in Parkinson’s Disease; A longitudinal intervention studyPLOS ONE

Dear Dr. Meijer,

Thank you for submitting your manuscript to PLOS ONE. After careful consideration, we feel that it has merit but does not fully meet PLOS ONE’s publication criteria as it currently stands. Therefore, we invite you to submit a revised version of the manuscript that addresses the points raised during the review process.

We look forward to receiving your revised manuscript.

Kind regards,

Julian Packheiser

Academic Editor

PLOS ONE

Additional Editor Comments:

Dear Larissa Meijer and colleagues,

attached you will find the reviewers' comments who have provided careful and constructive feedback on your manuscript. They acknowledge the merit and interest of the findings for the field of touch and I fully agree that this is a highly interesting study. They however also raised concerns that need to be addressed before the manuscript can become suitable for publication. During your revision, please pay attention especially towards the re-calculation of the results using a linear mixed model and a potential Bayesian counterpart to make more accurate statements about absence of evidence. Since the authors use SPSS where mixed modelling is possible in the frequentist domain, Bayesian models are not supported by that software. They can however be implemented using for example the brm function in R if the authors want to calculate a Bayesian model likewise. In the discussion, the limitation of the lack of a control group needs to be highlighted as potential pain attentuating effects could be for example attributed to temporal effects across the three weeks that are not associated with the treatment. While this is certainly less likely, other potential confounding factors need to be acknowledged and discussed due to a missing no touch control over the course of the experiment. As reviewer 2 notes, the graphical presentation of the study could be improved for easier accessibility of the results. Please keep in mind that showing raw data points in graphs is generally much more informative about data distributions rather than presenting means and SDs alone. I am looking forward to your revised manuscript!

Julian Packheiser

Reviewers' comments:

Reviewer's Responses to Questions

**Comments to the Author**

1. Is the manuscript technically sound, and do the data support the conclusions?

Reviewer #1: Partly

Reviewer #2: Partly

2. Has the statistical analysis been performed appropriately and rigorously? 

Reviewer #1: Yes

Reviewer #2: I Don't Know

3. Have the authors made all data underlying the findings in their manuscript fully available?

Reviewer #1: Yes

Reviewer #2: No

4. Is the manuscript presented in an intelligible fashion and written in standard English?

Reviewer #1: Yes

Reviewer #2: Yes

5. Review Comments to the Author

Reviewer #1: This study investigates if touch (slow and faster touch) can reduce chronic pain in patients with Parkinson’s Disease. Patients received both of these touch types in counterbalanced order from their partner. The main finding was that pain reports were lower following both touch types. Thus, a simple intervention provided by the partner at home can improve patients’ pain experience.

The research question is relevant and the manuscript is clearly written. The findings are of interest to the readers of PlosOne and to researchers in the field of touch and PD, and they also have clinical implications. However, there are shortcomings in the design and the analysis that need to be addressed before this manuscript can be recommended for publication.

Firstly, the design lacks a control group. This shortcoming is so far not discussed at all. Patient studies are rare and difficult to do, so the results can still be of interest, but the lack of a control group needs to be addressed. Secondly, whereas the data are carefully analysed, the type of analysis chosen is not optimal and should be modified. Thirdly, the authors claim to compare CT-optimal to CT-non-optimal touch by using stroking velocities of 3 and 18 cm/s. However, how CTs behave when being stroked with 18cm/s has never been investigated, so this term does not seem appropriate.

The authors should discuss the lack of a control group, change the analysis and the wording of “CT-non-optimal” throughout the text.

These and other points are further detailed here (comments in the order of the points in the manuscript they refer to):

1. Title and entire manuscript. The term “longitudinal” is somewhat misleading, as this term is typically used for interventions that run longer than just 3 weeks, often for years or even decades. I would suggest to remove that term to avoid confusion.

2. Abstract “patients underwent one week of pain registration, one week of affective touch and one week of non-affective touch”. This wording is unfortunate, as it give the impression that pain was only registered in the first week. Consider rewording “pain registration” to “baseline” or “no touch”.

3. Introduction, page 3, line 52. “affective touch (…) activates (….) C-Tactile afferent nerves”. Here you should cite a reference that actually measured whether gentle stroking activates CTs, i.e. one that used microneurography (e.g. Löken et al. Nat Neurosci 2009).

4. Materials and Methods, page 5. I suggest you move the points “Design” and “Procedure” at the beginning of the Materials and Measures section. It appears to be more logical to first provide the big picture, and then the details of the specific measures.

5. If possible, I also suggest to move the paragraph “Outline of data processing” from the Supplementary Materials into the methods. The paragraph “Statistical analysis” in the main text is not understandable without the paragraph “Outline of data processing”, and if it all were in the same place the reader would not have to scroll back and forth.

6. How was the sample size motivated? Can you provide a power calculation?

7. Quality of relationship, page 6, line 136 “discrepancy”, and Supplementary Materials Table S1, and Results page 10, line 246 “we assumed that the quality of their relationship did not influence touch perception”. I cannot find how this was analysed, but it seems that independent samples t-tests with the means of each group were calculated? With this method, you can arrive at exactly the same mean in both groups despite quite high differences in each pair. Thus, if the means are similar, it does not say anything about the discrepancy in each pair, and the conclusion that relationship quality not influence touch perception is not valid. If you are after that, you should calculate the differences for each pair and for example, correlate them to the pleasantness ratings.

8. Tactile stimulation, page 7, last paragraph. Please specify if the dorsal (or ventral) forearm was stroked.

9. Tactile stimulation, page 7, last paragraph. “CT non-optimal touch was administered by stroking (…) at (…) 18cm/s”. The velocity of 18 cm/s never been used in microneurography experiments that measured CT and Aβ activation. Thus, it is not known how CTs respond to velocities of 18 cm/s, and it is not known if 10 and 18 cm/s would give significantly different mean CT instantaneous firing frequencies. The wording “CT non-optimal” is not appropriate when it has never been tested. You could consider, for example, calling these velocities “slower” and “faster”.

10. Procedure. Figure 3 shows the big timeline of the intervention, but readers would probably appreciate a more fine-grained figure that also shows the single measurements (i.e., the time points at which the ratings are measured).

11. It is unclear if VAS pleasantness ratings were collected 3 or 4 times. Was there also a measurement before the very start of the 15 min period, i.e. at time point 0?

12. Statistical analysis, page 10. The authors performed a careful analysis of the data, but it does not seem optimal in several points. Firstly, Table 2 indicates that six t-tests are performed to compare faster and slower touch, and they seem not to have been corrected for multiple comparisons. Some correction for multiple comparison is required, and then the p-values will probably not survive the threshold. This would not make the findings less interesting. Thus, any number of t-tests calculated should be adjusted for multiple comparisons.

Secondly, there are three different repeated-measures-Anovas that can either be combined into one, or even better, be replaced by a linear mixed model (multilevel model). If a repeated measures Anova is calculated, it could be with the factors condition (no touch, fast, slow) and another factor timepoint (0, 5, 10, 15 minutes). This could nicely replace the different analyses for overall effect, short-term effect, and the one for “optimal duration”. Calculating the difference between timepoints as for the current “short-term effect” is then no longer required. If this approach is chosen, the factors and their levels should clearly be specified in the description of the analysis. This is currently not the case.

However, the better solution would be a linear mixed model, firstly because the information of the single trials would be used, rendering a higher power, and also because missing cases can be handled instead of the whole case being dropped as happens with repeated-measures Anova. I therefore ask the authors to calculate a linear mixed model with their data. For this, the single touch trials could be nested within one session (morning, afternoon). This would also allow to take into account the variability across stroking trials, which is interesting with respect to potential touch satiety.

13. Page 11, Table 1. For the reader it would be interesting to not only see the means, but also the data of each single subject. I suggest to move the table to the Supplements, and present a graph here where both the means and the individual data points are visible.

14. Page 14, line 315. It would be interesting to see (in a graph) how the pleasantness ratings change or do not change over time (i.e. 0,5,10, 15 min of stroking). Given the research on touch satiety (e.g. Triscoli et al. PLOS One 2014) and affective habituation (Dijksterhuis and Smith, Emotion 2002), one wonders if such a habituation process is abolished when stroking contributes to reduce pain. A closer look at this aspect may also help you further to disentangle pleasantness from pain effects. In particular, check out the paper by Taneja et al. (Front Pain Res 2021) who measured pain in addition to touch satiety.

15. Discussion, page 16 ff. The lack of a control condition needs to be discussed. The authors hint at the “social component underlying touch” (line 365), but need to make this aspect much more explicit. The current design of the study does not allow to state if the pain-relieving effects are due to touch itself or to what the authors call the “social component”. Since there was no control group, the results do not allow to conclude if the pain reduction has something to do with touch at all. Maybe the attention provided by the partner could be enough for such an effect, and pain would also be relieved if the partner painted the nails of the patient or performed a motivational interview? Note that touch by a partner conveys an intention of care, of being secure, and the availability of social support (Jakubiak and Feeney; Personality and Social Psychology Review 2017). Any other behaviour conveying similar intentions might have had the same effect. This does not reduce the importance of the current findings, but should be openly stated and reflected upon.

16. Page 17, lines 382-383 (“the pain ameliorating effect (…) is most effective between 10-15 min. If the authors find this interesting, they could relate this interval to other studies who investigated how the beneficial effects of slow touch evolve over time, for example heart rate (e.g. Püschel et al. Physiol Behav 2022; Triscoli et al., Biol Psychol 2017).

17. Page 18, first paragraph. It is very positive that the authors report how the patients commented the intervention. This adds important information. In this context, I wonder if the patients mentioned that they speak with each other during stroking? Do you have any information about that? To stroke or be stroked 15 minutes without saying anything appears rather awkward and boring for both parties. If the stroking was followed by (occasional) talking, this could on the one hand explain the overall high pleasantness ratings also for the faster touch, and maybe also have contributed to the pain-reducing effects of the intervention.

Reviewer #2: This longitudinal cross-over study tested the effects of affective touch from a partner on chronic pain in Parkison’s disease. Both affective and non-affective touch was effective in ameliorating pain on a day-to-day basis. In addition, affective touch had a stronger immediate (5-minute scale) on perceived pain.

This is a well-designed study on an important topic. I specifically applaud the effort to design the study in a way that allows for relatively direct application of the results as a clinical intervention.

Methods

- [major] One participant was excluded from the study “as the origin of his pain and study outcome were very different from the other participants in this study”. A more specific explanation is necessary. What was the exact difference between the origin of pain of the excluded participant, and which outcome parameters were different? Was this difference quantified in any way?

Results

- [major] Most data (except for the overall-effect) are presented in tables i.s.o. graphs. I think a more graphical presentation of the result would aid interpretation of the findings. I think this is especially important for the short-term effects, but also for example for the effects of (non-)affective touch on pleasantness scores. Preferably, these graphs would contain individual datapoints (or at least some information on the distributions).

- [major] There is a strong overlap between two analyses of the short-term effects. The first analysis consists of multiple paired t-tests for a touch-treatment effect on the baseline-corrected pain scores at different time points. The third analysis in this section is a repeated measures ANOVA of the raw pain scores with the factors time and touch-treatment (this analysis approach seems most comprehensive to me). The significant paired t-test for the 15min-0min pain score and the significant time x touch-treatment interaction seem to reflect the same underlying effect. I do not think it is appropriate to present them as entirely separate. I think the interaction effect is the most convincing result (a steeper slope for pain scores during affective touch).

- [minor] P. 17: “… the F-values are very low which indicates a low probability of a significant difference.” This suggests evidence for absence of an effect. Correct me if I am wrong, but I believe that this is not appropriate when using a frequentist approach (I think Bayesian modelling can be used to find evidence for the null hypothesis).

- [minor] The values for the “0min” touch in table 3 and table 4 differ. Should these values not be the same?

Discussion

- [major] The case report of the excluded participant (Meijer et al., 2023, journal of neuropsychology) ends with the remark that “further research in a larger clinical sample is warranted”. The current study seems like a prime example of such further research. I missed a discussion of how the current study relates to the case report (either explaining if the results of the current study support the previously reported findings, or explaining why such a comparison is not possible).

- [minor] The conclusion that CT-optimal touch is most effective after at least 10-15 minutes is too strong, given the available evidence: (1) Effects may appear earlier, but may just not have been detectable in the current study, given the relatively low sample size (2) It has not been tested if the pain reduction continues after the touch is finished. It is possible, for example, that a reduction in pain ratings still occurs at 15 minutes even if the touch is only applied for, say, 5 mins. This second point is relevant for the implementation of affective touch as a treatment.

- [minor] I was wondering if hormonal effects (e.g. oxytocin) may play a role in the potential underlying mechanisms, especially because the touch occurs between romantic partners.

- [minor] The discussion on conduction velocities (p. 23). The authors correctly mention that even though CT-fiber velocities are relatively slow (compared to A� fibers), they are still processed within seconds, whereas the (additive) effects of affective touch only occur in 10-15 minutes. However, the authors also seem to suggest that the fact that effects of affective touch occurs relatively late is somehow related to differences in velocities between A� fibers and CT-fibers. I cannot follow this reasoning.

6. PLOS authors have the option to publish the peer review history of their article (what does this mean?). If published, this will include your full peer review and any attached files.

Reviewer #1: No

Reviewer #2: No

---

## [Author Response · Author response to Decision Letter 0]

17 Nov 2023

Response to reviewers

This has been done. 

We have double checked the informed consent and ethical agreement and can inform you that we can share the fully anonymized data. This will be shared through YoDa, this is a public repository from Utrecht University. We can provide the DOI upon acceptance of the manuscript.

Additional Editor Comments:

Dear Larissa Meijer and colleagues,

attached you will find the reviewers' comments who have provided careful and constructive feedback on your manuscript. They acknowledge the merit and interest of the findings for the field of touch and I fully agree that this is a highly interesting study. They however also raised concerns that need to be addressed before the manuscript can become suitable for publication. During your revision, please pay attention especially towards the re-calculation of the results using a linear mixed model and a potential Bayesian counterpart to make more accurate statements about absence of evidence. Since the authors use SPSS where mixed modelling is possible in the frequentist domain, Bayesian models are not supported by that software. They can however be implemented using for example the brm function in R if the authors want to calculate a Bayesian model likewise. In the discussion, the limitation of the lack of a control group needs to be highlighted as potential pain attentuating effects could be for example attributed to temporal effects across the three weeks that are not associated with the treatment. While this is certainly less likely, other potential confounding factors need to be acknowledged and discussed due to a missing no touch control over the course of the experiment. As reviewer 2 notes, the graphical presentation of the study could be improved for easier accessibility of the results. Please keep in mind that showing raw data points in graphs is generally much more informative about data distributions rather than presenting means and SDs alone. I am looking forward to your revised manuscript!

Julian Packheiser

Dear Julian Packheiser,

First of all, thank you very much for the opportunity to revise our manuscript. In addition, we thank the editor en the reviewers for their constructive feedback and suggestions to improve the manuscript. Below, in red we addressed every point from the reviewers. We would also like to address some of the suggestions from the editor here. 

In regard to the data analysis, we looked into all possibilities to perform a linear mixed model as well as Bayesian Statistics. Unfortunately, it was not possible to perform these analyses due to the large and complex structure of the dataset. Therefore, we decided to perform 2 mixed model ANOVA’s as main analysis. Regarding the lack of a control group, as participants started with one week of only reporting their pain experience and this study was a within-subject design participants were their own control group. This was done because every individual experiences chronic pain differently, e.g. some have more pain in the evening others in the morning. Therefore, it was necessary to compare this within participants instead of only between. In addition, as pain needs to be present for at least 3 months to be classified as chronic pain it was not expected that the pain experience would change over time in such way that it is significantly lower at day 21 compared to day 1 of this study. However, we do agree that this is a limitation and therefore added this to the discussion. We created to new graphs in which individual datapoints as well as the mean are depicted. 

We hope that the changes made are sufficient and you would consider this manuscript for publication. 

Kind regards also on behalf of the co-authors,

Larissa Meijer 

Reviewers' comments:

Reviewer's Responses to Questions

Comments to the Author

1. Is the manuscript technically sound, and do the data support the conclusions?

Reviewer #1: Partly

Reviewer #2: Partly

2. Has the statistical analysis been performed appropriately and rigorously? 

Reviewer #1: Yes

Reviewer #2: I Don't Know

3. Have the authors made all data underlying the findings in their manuscript fully available?

Reviewer #1: Yes

Reviewer #2: No

4. Is the manuscript presented in an intelligible fashion and written in standard English?

Reviewer #1: Yes

Reviewer #2: Yes

5. Review Comments to the Author

Reviewer #1: This study investigates if touch (slow and faster touch) can reduce chronic pain in patients with Parkinson’s Disease. Patients received both of these touch types in counterbalanced order from their partner. The main finding was that pain reports were lower following both touch types. Thus, a simple intervention provided by the partner at home can improve patients’ pain experience.

The research question is relevant and the manuscript is clearly written. The findings are of interest to the readers of PlosOne and to researchers in the field of touch and PD, and they also have clinical implications. However, there are shortcomings in the design and the analysis that need to be addressed before this manuscript can be recommended for publication.

Firstly, the design lacks a control group. This shortcoming is so far not discussed at all. Patient studies are rare and difficult to do, so the results can still be of interest, but the lack of a control group needs to be addressed. Secondly, whereas the data are carefully analysed, the type of analysis chosen is not optimal and should be modified. Thirdly, the authors claim to compare CT-optimal to CT-non-optimal touch by using stroking velocities of 3 and 18 cm/s. However, how CTs behave when being stroked with 18cm/s has never been investigated, so this term does not seem appropriate.

The authors should discuss the lack of a control group, change the analysis and the wording of “CT-non-optimal” throughout the text.

These and other points are further detailed here (comments in the order of the points in the manuscript they refer to):

1. Title and entire manuscript. The term “longitudinal” is somewhat misleading, as this term is typically used for interventions that run longer than just 3 weeks, often for years or even decades. I would suggest to remove that term to avoid confusion.

We removed the term from the title and also changed this within the manuscript. 

2. Abstract “patients underwent one week of pain registration, one week of affective touch and one week of non-affective touch”. This wording is unfortunate, as it give the impression that pain was only registered in the first week. Consider rewording “pain registration” to “baseline” or “no touch”.

We changed this sentence accordingly. Line number 19-20.

3. Introduction, page 3, line 52. “affective touch (…) activates (….) C-Tactile afferent nerves”. Here you should cite a reference that actually measured whether gentle stroking activates CTs, i.e. one that used microneurography (e.g. Löken et al. Nat Neurosci 2009).

We changed the reference accordingly, line 57.

4. Materials and Methods, page 5. I suggest you move the points “Design” and “Procedure” at the beginning of the Materials and Measures section. It appears to be more logical to first provide the big picture, and then the details of the specific measures.

We placed the Design and Procedure sections below the Participants section. From line 129 – 176. 

5. If possible, I also suggest to move the paragraph “Outline of data processing” from the Supplementary Materials into the methods. The paragraph “Statistical analysis” in the main text is not understandable without the paragraph “Outline of data processing”, and if it all were in the same place the reader would not have to scroll back and forth.

Outline of data processing has been added to the Statistical analysis paragraph. It has therefore een removed from the Supplementary materials. Line numbers 283 – 335. 

6. How was the sample size motivated? Can you provide a power calculation?

The apriori sample size calculation has been added to the method section (line numbers 290 - 293), as well as the observed power of the main analysis (.995) which shows that there is high statistical power for N=17. 

7. Quality of relationship, page 6, line 136 “discrepancy”, and Supplementary Materials Table S1, and Results page 10, line 246 “we assumed that the quality of their relationship did not influence touch perception”. I cannot find how this was analysed, but it seems that independent samples t-tests with the means of each group were calculated? With this method, you can arrive at exactly the same mean in both groups despite quite high differences in each pair. Thus, if the means are similar, it does not say anything about the discrepancy in each pair, and the conclusion that relationship quality not influence touch perception is not valid. If you are after that, you should calculate the differences for each pair and for example, correlate them to the pleasantness ratings.

We replaced the t-test with a Spearman correlation between the QRI difference score and VAS pleasantness for CT-optimal touch and CT non-optimal touch. This is described in the result section line numbers 476 – 481. 

8. Tactile stimulation, page 7, last paragraph. Please specify if the dorsal (or ventral) forearm was stroked.

This was the dorsal part (hairy skin), this has been specified at line number 233. 

9. Tactile stimulation, page 7, last paragraph. “CT non-optimal touch was administered by stroking (…) at (…) 18cm/s”. The velocity of 18 cm/s never been used in microneurography experiments that measured CT and Aβ activation. Thus, it is not known how CTs respond to velocities of 18 cm/s, and it is not known if 10 and 18 cm/s would give significantly different mean CT instantaneous firing frequencies. The wording “CT non-optimal” is not appropriate when it has never been tested. You could consider, for example, calling these velocities “slower” and “faster”.

We agree with the reviewer that the velocity of 18 cm/s has not been studied specifically in microneurography experiments. However, we do know from these studies that the optimal velocity for CT-optimal touch is 3 cm/s with a range of 1 – 10 cm/s and that when faster stimuli >10cm/s are applied that only a handful of spike with lower frequency respond compared to optimal velocity 1 -10cm/s (Ackerley, Sci Rep 2022) Therefore, we consider 18 cm/s non-optimal as this is faster than the 10 cm/s which is considered a CT-optimal velocity. In addition to this, this term has been used before by other others as well (see von Mohr et al Soc Cog & Aff Neurosc 2018) but also in our own previous studies (see Meijer et al JNP 2022, Meijer et al Sci Rep 2022). 

Taken together, we would prefer to keep the term CT non-optimal touch to stay in line with previous studies. As we do agree that we do not know exactly how the CT-fibers react to 18 cm/s we added this in the discussion section line numbers 515 - 519. 

10. Procedure. Figure 3 shows the big timeline of the intervention, but readers would probably appreciate a more fine-grained figure that also shows the single measurements (i.e., the time points at which the ratings are measured).

This has been added to Figure 3. 

11. It is unclear if VAS pleasantness ratings were collected 3 or 4 times. Was there also a measurement before the very start of the 15 min period, i.e. at time point 0?

There were 3 measurements at 5, 10, 15 min so after every 5min of touch administration. This has been clarified in the Procedure section line number 167 – 168. 

12. Statistical analysis, page 10. The authors performed a careful analysis of the data, but it does not seem optimal in several points. Firstly, Table 2 indicates that six t-tests are performed to compare faster and slower touch, and they seem not to have been corrected for multiple comparisons. Some correction for multiple comparison is required, and then the p-values will probably not survive the threshold. This would not make the findings less interesting. Thus, any number of t-tests calculated should be adjusted for multiple comparisons.

Secondly, there are three different repeated-measures-Anovas that can either be combined into one, or even better, be replaced by a linear mixed model (multilevel model). If a repeated measures Anova is calculated, it could be with the factors condition (no touch, fast, slow) and another factor timepoint (0, 5, 10, 15 minutes). This could nicely replace the different analyses for overall effect, short-term effect, and the one for “optimal duration”. Calculating the difference between timepoints as for the current “short-term effect” is then no longer required. If this approach is chosen, the factors and their levels should clearly be specified in the description of the analysis. This is currently not the case.

However, the better solution would be a linear mixed model, firstly because the information of the single trials would be used, rendering a higher power, and also because missing cases can be handled instead of the whole case being dropped as happens with repeated-measures Anova. I therefore ask the authors to calculate a linear mixed model with their data. For this, the single touch trials could be nested within one session (morning, afternoon). This would also allow to take into account the variability across stroking trials, which is interesting with respect to potential touch satiety.

We agree that a linear mixed model would be the better solution. We tried performing a linear mixed model but unfortunately this was not possible. SPSS as well as JASP could not run the analysis as there were too many levels and factors. Even when we nested trials in different categories or used average FPS-R scores, we received a warning that parameters could not be estimated and that we needed to reduce the random effects structure. This was not possible. 

As such we decided to run two repeated measures ANOVA. We could not use one repeated measures ANOVA for all data, as suggested, because the factor timepoint is not included in the no touch condition. As we agree that we lose a lot of variability by calculating week averages, we decided to include the factor days into the analysis. As such one participant was excluded from the analysis as there was too much missing data possibly invalidating the individual data as well. This has been changed in the result section line numbers 367 – 442. 

13. Page 11, Table 1. For the reader it would be interesting to not only see the means, but also the data of each single subject. I suggest to move the table to the Supplements, and present a graph here where both the means and the individual data points are visible.

We made a new graph with individual data points see Figure 4. Here we calculated FPS-R difference scores for CT-optimal touch – no touch and CT non-optimal touch – no touch. This was done for every participant and every day of the week. 

14. Page 14, line 315. It would be interesting to see (in a graph) how the pleasantness ratings change or do not change over time (i.e. 0,5,10, 15 min of stroking). Given the research on touch satiety (e.g. Triscoli et al. PLOS One 2014) and affective habituation (Dijksterhuis and Smith, Emotion 2002), one wonders if such a habituation process is abolished when stroking contributes to reduce pain. A closer look at this aspect may also help you further to disentangle pleasantness from pain effects. In particular, check out the paper by Taneja et al. (Front Pain Res 2021) who measured pain in addition to touch satiety.

To illustrate the difference between 5, 10, 15min we calculated the average per individual for these different timepoints, as shown in the graph this is stable per individual. Furthermore, the pleasantness analysis has been changed to a mixed Anova including the factor day as well. As there is no interaction between touch x day it appears that touch perception is stable over time. We did not include this in the manuscript for now, as the manuscript is already dense but if necessary, we could add this in the supplementary material file. 

15. Discussion, page 16 ff. The lack of a control condition needs to be discussed. The authors hint at the “social component underlying touch” (line 365), but need to make this aspect much more explicit. The current design of the study does not allow to state if the pain-relieving effects are due to touch itself or to what the authors call the “social component”. Since there was no control group, the results do not allow to conclude if the pain reduction has something to do with touch at all. Maybe the attention provided by the partner could be enough for such an effect, and pain would also be relieved if the partner painted the nails of the patient or performed a motivational interview? Note that touch by a partner conveys an intention of care, of being secure, and the availability of social support (Jakubiak and Feeney; Personality and Social Psychology Review 2017). Any other behaviour conveying similar intentions might have had the same effect. This does not reduce the importance of the current findings, but should be openly stated and reflected upon.

We agree that this is an important limitation, this has been added to the discussion at line numbers 634 – 645. 

16. Page 17, lines 382-383 (“the pain ameliorating effect (…) is most effective between 10-15 min. If the authors find this interesting, they could relate this interval to other studies who investigated how the beneficial effects of slow touch evolve over time, for example heart rate (e.g. Püschel et al. Physiol Behav 2022; Triscoli et al., Biol Psychol 2017).

We decided to only mention this briefly as there are several things we still do not know regarding touch duration and therefore do not want to make strong statements regarding this point. See line numbers 529 – 533. 

17. Page 18, first paragraph. It is very positive that the authors report how the patients commented the intervention. This adds important information. In this context, I wonder if the patients mentioned that they speak with each other during stroking? Do you have any information about that? To stroke or be stroked 15 minutes without saying anything appears rather awkward and boring for both parties. If the stroking was followed by (occasional) talking, this could on the one hand explain the overall high pleasantness ratings also for the faster touch, and maybe also have contributed to the pain-reducing effects of the intervention.

We asked the participants to perform touch in a quite environment with no distractions and also to not talk and really be in the moment and have their attention fully on the touch experience. This is something we could not control for of course because we were not present during the touch administration. However, participants and their partners subjectively reported that for them it was not awkward or strange to be quite as they felt really connected and in the moment when they were having 15 min with just the two of them. 

Reviewer #2: This longitudinal cross-over study tested the effects of affective touch from a partner on chronic pain in Parkison’s disease. Both affective and non-affective touch was effective in ameliorating pain on a day-to-day basis. In addition, affective touch had a stronger immediate (5-minute scale) on perceived pain.

This is a well-designed study on an important topic. I specifically applaud the effort to design the study in a way that allows for relatively direct application of the results as a clinical intervention.

Methods

- [major] One participant was excluded from the study “as the origin of his pain and study outcome were very different from the other participants in this study”. A more specific explanation is necessary. What was the exact difference between the origin of pain of the excluded participant, and which outcome parameters were different? Was this difference quantified in any way?

We added more information regarding the exclusion of this particular case from the larger sample. See line numbers 117 – 125. 

Results

- [major] Most data (except for the overall-effect) are presented in tables i.s.o. graphs. I think a more graphical presentation of the result would aid interpretation of the findings. I think this is especially important for the short-term effects, but also for example for the effects of (non-)affective touch on pleasantness scores. Preferably, these graphs would contain individual datapoints (or at least some information on the distributions).

We made a new graph with individual data points see Figure 4. Here we calculated FPS-R difference scores for CT-optimal touch – no touch and CT non-optimal touch – no touch. This was done for every participant and every day of the week.

- [major] There is a strong overlap between two analyses of the short-term effects. The first analysis consists of multiple paired t-tests for a touch-treatment effect on the baseline-corrected pain scores at different time points. The third analysis in this section is a repeated measures ANOVA of the raw pain scores with the factors time and touch-treatment (this analysis approach seems most comprehensive to me). The significant paired t-test for the 15min-0min pain score and the significant time x touch-treatment interaction seem to reflect the same underlying effect. I do not think it is appropriate to present them as entirely separate. I think the interaction effect is the most convincing result (a steeper slope for pain scores during affective touch).

We agree with this important point therefore we decided to only report the repeated measures ANOVA in which we included the factor days instead of only looking at the data over the week. Here we find an interaction effect between touch x timepoint in which there is indeed a steeper slope for CT-optimal touch compared to CT non-optimal touch. This is also visualized in Figure 5. See the result section line numbers 393 – 442. 

- [minor] P. 17: “… the F-values are very low which indicates a low probability of a significant difference.” This suggests evidence for absence of an effect. Correct me if I am wrong, but I believe that this is not appropriate when using a frequentist approach (I think Bayesian modelling can be used to find evidence for the null hypothesis). 

We removed this sentence. In addition, we also tried Bayesian modelling on the overall and short-term effect. However, for the short-term effect the error % were around 99 reflecting an inaccuracy of the model, which is likely related to the large and complex dataset. Therefore we decided to not include Bayesian modelling. 

- [minor] The values for the “0min” touch in table 3 and table 4 differ. Should these values not be the same?

We removed Table 3, only Table 4 is included in the text. 

Discussion

- [major] The case report of the excluded participant (Meijer et al., 2023, journal of neuropsychology) ends with the remark that “further research in a larger clinical sample is warranted”. The current study seems like a prime example of such further research. I missed a discussion of how the current study relates to the case report (either explaining if the results of the current study support the previously reported findings, or explaining why such a comparison is not possible).

We wrote a new paragraph on this comparison within the discussion section. Line numbers 602 – 619. 

- [minor] The conclusion that CT-optimal touch is most effective after at least 10-15 minutes is too strong, given the available evidence: (1) Effects may appear earlier, but may just not have been detectable in the current study, given the relatively low sample size (2) It has not been tested if the pain reduction continues after the touch is finished. It is possible, for example, that a reduction in pain ratings still occurs at 15 minutes even if the touch is only applied for, say, 5 mins. This second point is relevant for the implementation of affective touch as a treatment.

This is indeed a relevant point, which we added to the discussion. We also changed this paragraph a little to make the statement less strong. Line numbers 520 – 533.

- [minor] I was wondering if hormonal effects (e.g. oxytocin) may play a role in the potential underlying mechanisms, especially because the touch occurs between romantic partners.

This is an interesting point that warrants further investigation. However, we believe this is beyond the scope of the current study.

- [minor] The discussion on conduction velocities (p. 23). The authors correctly mention that even though CT-fiber velocities are relatively slow (compared to A� fibers), they are still processed within seconds, whereas the (additive) effects of affective touch only occur in 10-15 minutes. However, the authors also seem to suggest that the fact that effects of affective touch occurs relatively late is somehow related to differences in velocities between A� fibers and CT-fibers. I cannot follow this reasoning.

We believed this point made things unclear and was highly speculative as we did not use microneurography. Therefore we removed this part. 

---

## [Decision Letter · Decision Letter 1]

18 Dec 2023

PONE-D-23-21643R1CT-optimal touch and chronic pain experience in Parkinson’s Disease; An intervention studyPLOS ONE

Dear Dr. Meijer,

Thank you for submitting your manuscript to PLOS ONE. After careful consideration, we feel that it has merit but does not fully meet PLOS ONE’s publication criteria as it currently stands. Therefore, we invite you to submit a revised version of the manuscript that addresses the points raised during the review process.

We look forward to receiving your revised manuscript.

Kind regards,

Julian Packheiser

Academic Editor

PLOS ONE

Journal Requirements:

Reviewers' comments:

Reviewer's Responses to Questions

**Comments to the Author**

1. If the authors have adequately addressed your comments raised in a previous round of review and you feel that this manuscript is now acceptable for publication, you may indicate that here to bypass the “Comments to the Author” section, enter your conflict of interest statement in the “Confidential to Editor” section, and submit your "Accept" recommendation.

Reviewer #1: (No Response)

Reviewer #2: All comments have been addressed

2. Is the manuscript technically sound, and do the data support the conclusions?

Reviewer #1: Partly

Reviewer #2: Yes

3. Has the statistical analysis been performed appropriately and rigorously? 

Reviewer #1: Yes

Reviewer #2: Yes

4. Have the authors made all data underlying the findings in their manuscript fully available?

Reviewer #1: Yes

Reviewer #2: No

5. Is the manuscript presented in an intelligible fashion and written in standard English?

Reviewer #1: Yes

Reviewer #2: Yes

6. Review Comments to the Author

Reviewer #1: The authors present a fine revision of their manuscript and have well answered to my questions and comments. I only have a few minor suggestions and a more important one for interpreting the results.

Abstract: I think it should read “treated with analgesics” (not analgesia)

Line 67: “appreciation” means recognizing the good qualities of something (Oxford dictionary). I don’t think this is what is meant here.

Line 108: explain what “Gorilla” is when it is mentioned for the first time.

Figure 4 and 5: Say in the caption what the line shows (mean values I assume).

Line 329. There is no “S2 Table”. Do you mean “Table 2”?

Line 380 ff. “an immediate effect of touch on chronic pain experience was also investigated. Here, the results showed that CT-optimal touch reduces the chronic pain experience significantly more compared to CT non-optimal touch”. This appears to be the interpretation of the results in lines 324-329. However, this interpretation does not seem justified. Similar to a pain reduction of CT-optimal touch, namely a significant difference between 0 and 5 minutes, and between 0 and 10, there is also a significant difference for CT-non-optimal touch between 0 and 5 minutes, and between 0 and 10. Thus, there is also a short-time pain-reducing effect of CT-non-optimal touch.

The only difference between CT-optimal and non-optimal touch is that for CT-optimal touch, there is an additional significant difference for the time interval between 5 and 10 minutes, which is not present for CT-non-optimal touch. For the other four intervals, CT and non-CT touch yield similar results. Thus, the interpretation that CT-touch is more efficient than non-CT-touch for short-term pain relief is on shaky grounds as it is based on just this one difference (one out of five). Consequently, it should be amended in both the discussion and the abstract. The conclusion appears to rather be that CT and non-CT touch are equally effective in short- as well as long-term pain reduction, which is perfectly fine. This would also fit to the findings from studies showing that even mere partner presence without touch can reduce pain (e.g.; Duschek et al, Scand J Pain 2019). I suggest you consider that it might be the support communicated through touch (or other means) that is crucial for pain amelioration, and not the type of touch itself. I leave this to the authors, but it might be interesting to discuss this aspect. Personally, I think this would be more beneficial to the reader than exaggerating the small CT effects.

Line 481 “So, it could be that any form of affectionate touch might have similar effects”. This sentence is not understandable unless the authors want to say that only touch that is performed with CT-optimal velocity is affectionate? This claim seems to not be justified since other forms of touch such as hugs and handholding can certainly be perceived as affectionate. Do the authors mean “Any form of affectionate behaviour”, e.g. verbal support? This would make sense. Please reformulate.

Line 482 “as there is an additional relieving effect of CT-optimal touch”. This is an overstatement, see comment further up. I would suggest to remove this sentence.

Reviewer #2: All my coments have been addressed and the report has definitely been improved.

For future reference: the line numbers in the response letter did not match the manuscript (if I'm correct). I also think that the language can be improved here and there.

7. PLOS authors have the option to publish the peer review history of their article (what does this mean?). If published, this will include your full peer review and any attached files.

Reviewer #1: No

Reviewer #2: No

---

## [Author Response · Author response to Decision Letter 1]

22 Dec 2023

Response to reviewers

Dear Julian Packheiser,

First of all, we would like to thank the editor and reviewers for the opportunity to address these minor revisions. These extra valuable comments have improved the manuscript even more. We sincerely hope that it now meets the criteria for publication in PLOSone. Below, in red we addressed every point from the reviewers. The line numbers refer to the manuscript with track changes.

Thank you again for considering our manuscript and we would like to wish you a Merry Christmas and a Happy New Year. 

Kind regards also on behalf of the co-authors,

Larissa Meijer 

Journal Requirements:

The reference list has been checked and is correct, no adjustments have been made. 

Reviewers' comments:

Reviewer's Responses to Questions 

Comments to the Author

1. If the authors have adequately addressed your comments raised in a previous round of review and you feel that this manuscript is now acceptable for publication, you may indicate that here to bypass the “Comments to the Author” section, enter your conflict of interest statement in the “Confidential to Editor” section, and submit your "Accept" recommendation.

Reviewer #1: (No Response)

Reviewer #2: All comments have been addressed

2. Is the manuscript technically sound, and do the data support the conclusions?

Reviewer #1: Partly

Reviewer #2: Yes

3. Has the statistical analysis been performed appropriately and rigorously? 

Reviewer #1: Yes

Reviewer #2: Yes

4. Have the authors made all data underlying the findings in their manuscript fully available?

Reviewer #1: Yes

Reviewer #2: No

5. Is the manuscript presented in an intelligible fashion and written in standard English?

Reviewer #1: Yes

Reviewer #2: Yes

6. Review Comments to the Author

Reviewer #1: The authors present a fine revision of their manuscript and have well answered to my questions and comments. I only have a few minor suggestions and a more important one for interpreting the results.

Abstract: I think it should read “treated with analgesics” (not analgesia)

This is indeed wrong, we changed it to analgesics at line number 13.

Line 67: “appreciation” means recognizing the good qualities of something (Oxford dictionary). I don’t think this is what is meant here.

We changed this sentence to ‘which are important for the affective aspects of pain experience’ at line numbers 61-62. 

Line 108: explain what “Gorilla” is when it is mentioned for the first time.

We added the following ‘’ Gorilla (an online survey tool)… “ at line number 110. 

Figure 4 and 5: Say in the caption what the line shows (mean values I assume).

This has been added to the figure captions (see Figure file). 

Line 329. There is no “S2 Table”. Do you mean “Table 2”?

We were referring to the S2 Table. I have checked the revised supplementary File as a supporting information file, the S2 Table is available there. This has been done according to the submission guidelines from PLOSone. 

Line 380 ff. “an immediate effect of touch on chronic pain experience was also investigated. Here, the results showed that CT-optimal touch reduces the chronic pain experience significantly more compared to CT non-optimal touch”. This appears to be the interpretation of the results in lines 324-329. However, this interpretation does not seem justified. Similar to a pain reduction of CT-optimal touch, namely a significant difference between 0 and 5 minutes, and between 0 and 10, there is also a significant difference for CT-non-optimal touch between 0 and 5 minutes, and between 0 and 10. Thus, there is also a short-time pain-reducing effect of CT-non-optimal touch.

The only difference between CT-optimal and non-optimal touch is that for CT-optimal touch, there is an additional significant difference for the time interval between 5 and 10 minutes, which is not present for CT-non-optimal touch. For the other four intervals, CT and non-CT touch yield similar results. Thus, the interpretation that CT-touch is more efficient than non-CT-touch for short-term pain relief is on shaky grounds as it is based on just this one difference (one out of five). Consequently, it should be amended in both the discussion and the abstract. The conclusion appears to rather be that CT and non-CT touch are equally effective in short- as well as long-term pain reduction, which is perfectly fine. This would also fit to the findings from studies showing that even mere partner presence without touch can reduce pain (e.g.; Duschek et al, Scand J Pain 2019). I suggest you consider that it might be the support communicated through touch (or other means) that is crucial for pain amelioration, and not the type of touch itself. I leave this to the authors, but it might be interesting to discuss this aspect. Personally, I think this would be more beneficial to the reader than exaggerating the small CT effects.

We fully agree that this is effect is overstated. These statements were based on the previously reported t-test. As we removed this analysis, these statements should be adjusted as well. Based on the interaction effect between touchxtimepoint and the steeper slope of CT-optimal touch in Fig 5, it appears that it is slightly more beneficial to use CT-optimal touch at the 15min timepoint compared to CT non-optimal touch. As it was also perceived as more pleasant and more feasible to apply for the partners, we do believe CT-optimal touch would be the preferable choice compared to CT non-optimal touch. We also added some information regarding the mere presence of partner as a form of passive social support at line numbers 508-509. 

However, as mentioned this was still overstated so we changed this throughout the manuscript at line numbers: 21-22, 331-332, 390-392, 413-416, 476-486 and 522-526.

Line 481 “So, it could be that any form of affectionate touch might have similar effects”. This sentence is not understandable unless the authors want to say that only touch that is performed with CT-optimal velocity is affectionate? This claim seems to not be justified since other forms of touch such as hugs and handholding can certainly be perceived as affectionate. Do the authors mean “Any form of affectionate behaviour”, e.g. verbal support? This would make sense. Please reformulate.

As we indeed wanted to make this exact point, we reformulated this to affectionate behaviour at line number 507. 

Line 482 “as there is an additional relieving effect of CT-optimal touch”. This is an overstatement, see comment further up. I would suggest to remove this sentence.

We removed this sentence and were more cautious with the other statements regarding the additional effect of CT-optimal touch. Also see the changes mentioned above. 

Reviewer #2: All my coments have been addressed and the report has definitely been improved.

For future reference: the line numbers in the response letter did not match the manuscript (if I'm correct). I also think that the language can be improved here and there.

7. PLOS authors have the option to publish the peer review history of their article (what does this mean?). If published, this will include your full peer review and any attached files.

Do you want your identity to be public for this peer review? For information about this choice, including consent withdrawal, please see our Privacy Policy.

Reviewer #1: No

Reviewer #2: No

---

## [Editor Report · Decision Letter 2]

23 Jan 2024

CT-optimal touch and chronic pain experience in Parkinson’s Disease; An intervention study

PONE-D-23-21643R2

Dear Dr. Meijer,

We’re pleased to inform you that your manuscript has been judged scientifically suitable for publication and will be formally accepted for publication once it meets all outstanding technical requirements.

Kind regards,

Julian Packheiser

Academic Editor

PLOS ONE
---

## [Editor Report · Acceptance letter]

14 Feb 2024

PONE-D-23-21643R2 

PLOS ONE

Dear Dr. Meijer, 

I'm pleased to inform you that your manuscript has been deemed suitable for publication in PLOS ONE. Congratulations! Your manuscript is now being handed over to our production team.

Kind regards, 

on behalf of

Dr. Julian Packheiser 

Academic Editor

PLOS ONE